# Host plant nutrition drives fitness outcomes in the cactus specialist *Drosophila mettleri*

**Lidane Noronha**[ID]◉*, **Brian P. Lazzaro**[ID]◉, **Patrick M. O'Grady**◉

Department of Entomology, Cornell University, Ithaca, New York, United States of America

◉ These authors contributed equally to this work.
* lcn27@cornell.edu

## Abstract

Organisms must navigate complex interactions with host plants, microbial communities, and environmental cues to ensure their survival and reproductive success when adapting to novel environments. Due to their ecological constraints, host plant specialists can be used to study how these interactions affect fitness due to their ecological constraints. In specialist species, such as cactophilic *Drosophila*, it remains unclear how feeding behavior, substrate composition, and microbial interactions collectively shape fitness outcomes. We examined the effects of laboratory media (cornmeal vs. banana) differing in their base diet and cactus-derived additives (dried Saguaro powder, exudate from rotting Saguaro, soil soaked by rotting Saguaro) on fitness in *Drosophila mettleri*, a columnar cactus specialist that breeds in Saguaro cactus (*Carnegiea gigantea*) in the Sonoran Desert. Cactus supplements often reduced survival from egg to pupa, but increased survival from pupa to adult, resulting in stage specific tradeoffs shaping egg-to-adult fitness. Results show interactions between food substrate and cactus treatment: cactus supplementation reduced survival on banana media but increased survival on cornmeal-based diets. Feeding rate and overall amount of media consumed did not differ among treatments, indicating that differences in survival and fitness may depend on the broader nutritional environment and developmental stage. This suggests that studying host specialization should include multiple life stages.

## Introduction

Adapting to a novel host plant may require several steps, including tolerance to plant defensive compounds, developing a preference for the volatile and nutritional profile of a given host substrate, and subsequently becoming dependent on that host for metabolism and development [1,2]. Host shifts can affect oviposition behavior, morphology, interspecific competition, and, at times, can lead to specialization [3–5]. Tripartite symbioses between insects, plants, and microbes, such as the cactus-yeast-*Drosophila* system, has been used to examine the broader effects of host

**Data availability statement:** The dataset generated during the current study is available in the Dryad Digital Repository (https://doi.org/10.5061/dryad.x3ffbg7z6).

**Funding:** This paper was supported by National Science Foundation (https://www.nsf.gov/bio/deb) grants DEB2030129, DEB1839598, and DEB1241253 to PMO. The Cornell University Insect Collection (https://cuic.entomology.cornell.edu/) and the American Museum of Natural History Theodore Roosevelt Memorial Grant (https://www.amnh.org/research/richard-gilder-graduate-school/grants) provided financial support to LN. The funders had no role in study design, data collection and analysis, decision to publish, or preparation of the manuscript.

**Competing interests:** The authors have declared that no competing interests exist.

plant specialization [5–10]. However, little is known about how cactus and yeast act together to affect the fitness of *Drosophila* species.

Deserts exist on the edge of environmental extremes, making them a natural laboratory to study host shifts under climatic pressure [11]. The added layer of environmental stresses, such as scarcity of water and food sources, can lead to close evolutionary relationships between insects and their host plants, with microbes mediating these interactions [11–13]. The Saguaro cactus, *Carnegiea gigantea*, is emblematic of the Sonoran Desert [14]. Saguaro produces alkaloid toxins that are toxic to most animals, except the few that have evolved mechanisms to detoxify the exudate [15,16].

Two species of the *Drosophila repleta* group present in the Sonoran Desert, *Drosophila nigrospiracula* and *Drosophila mettleri*, are commonly associated with necrotic *C. gigantea*, though they utilize distinct microhabitats within this environment [10,17]. Although *D. mettleri* is often associated with Saguaro in the Sonoran Desert, this species also occurs in regions lacking Saguaro. *D. mettleri* oviposits in soil soaked with necrotic cactus tissue [18]. High evaporation rates within the arid Sonoran Desert concentrate toxins in the soil as the exudate accumulates [19], rendering this environment free of competition from other arthropods, including the closely related D. nigrospiracula, which oviposits in the necrotic Saguaro cactus tissue [6,10].

Many members of the cactophilic *Drosophila repleta* species group can be reared under laboratory conditions without their host plant material [9,20,21]. *D. mettleri* is an exception, as successful laboratory culture typically requires the incorporation of cactus-derived material. This dependence may reflect nutritional requirements of the host substrate [15–17,21,22], contributions from microbes associated with cactus [7,8,17], or their interaction. Recently, several studies have suggested that Saguaro may provide an olfactory or gustatory cue that stimulates *Drosophila* feeding and/or oviposition, leading to a successful laboratory culture [23,24]. Previous studies using other cactophilic *Drosophila* species suggest that increased developmental success is correlated to nutrient availability, with a preference for media containing added cactus [22,25–29]. The process of host plant specialization can be studied by examining how Saguaro and the microbes within the decomposing cacti impact the fitness of *D. mettleri* [3,5].

This study explores the behavioral and physiological effects of diet in *Drosophila mettleri* and uses these results to infer the long-term fitness implications of specialization on the Saguaro cactus [30–32]. We reared *D. mettleri* adults individually on cornmeal and banana media to test whether experimental trials with added Saguaro would improve *D. mettleri* pupation rate compared to control diets lacking Saguaro. We compared cactus additives that differed in microbial abundance to assess how their interactions with base dietary composition affects fitness. We also considered a potential interaction with natural microbes, where the diet might either mitigate any negative effects of microbes or where the microorganisms could enhance nutrient availability from the diet [16,32–35]. Feeding was quantified to rule out differences in consumption as a confounding factor. By integrating stage-specific survival with

egg-to-adult fitness, we aim to clarify how host-derived substrates interact with laboratory diets to shape developmental success in a specialist insect.

## Materials and methods

### *Drosophila* and field sample collection

A laboratory culture of *Drosophila mettleri* (15081–1502.15) was obtained from the National *Drosophila* Species Stock Center. This line was collected in La Paz, Baja California Sur, Mexico in 2012, where its primary host cactus is cardón (*Pachycereus pringlei*). The line has been maintained on Banana media and Saguaro powder since 2012 [36], and the long-term laboratory maintenance of this line is noted as a caveat, as it may have undergone adaptation to laboratory conditions. *Drosophila mettleri* was used for Nutritional Assays and Excrement Quantification experiments. Powdered Saguaro was maintained at −4°C in 50mL Falcon tubes. Exudate from necrotic Saguaro arms and soil collected from the base of these necrotic tissues were collected from an actively rotting Saguaro near Globe, Arizona, USA 33.57635°N, 110.95942°W. These were scraped into snap caps and stored at 21°C. For each treatment, cactus powder, exudate, or soil was homogenized and dispensed into vials at a standardized 0.14% (w/v). This study involved only invertebrate species (*Drosophila mettleri*) and therefore did not require approval by an Institutional Animal Care and Use Committee (IACUC), as confirmed by the Cornell University IACUC Office. We obtained permits from Saguaro National Parks permit number SAGU-2023-SCI-0007 to conduct some aspects of the field work.

### Fitness assay

The Fitness Assay was conducted to directly compare cactus additives of varying microbial activity. We tested eight diets using either Banana or Cornmeal media as a base with one of four additive treatments: a negative control where no additives are added, and three experimental trials with various additives (Saguaro powder from the NDSSC, Saguaro exudate from Arizona, and soil soaked with Saguaro necrosis from Arizona) (Fig 1). The banana media [37] consisted of 6% (w/v) sugar, 11% (w/v) protein, and 16% (w/v) carbohydrate. The cornmeal media [38] consists of 4% (w/v) sugar, 9% (w/v) protein, and 15% (w/v) carbohydrate. The *Drosophila* Dietary Composition Calculator (DDCC) [34] was developed based on previously published *Drosophila* media recipes and used to determine the exact nutrient profiles of our recipes. Each treatment had five replicates with five mating pairs in each vial (400 adult flies in total). Male and female adult flies were separately aged for 48 hours on specific treatments, then combined into vials based on their assigned treatment. Flies were held in experimental vials for 24 hours. After this time, they were serially transferred every 24 hours, and any eggs laid within each 24 hour interval were counted.

Adult transfer was repeated daily for five days, after which the adults were preserved in ethanol. Eggs were counted with a clicker counter and by marking the starting point, counting eggs in sight while turning clockwise. After reaching the starting point, eggs in the middle are counted from left to right using obvious markers to avoid overcounting eggs. Dates of the first larvae and first pupae per vial were noted. Larval stages could not be accurately distinguished in this rearing setup, so they were not recorded. Pupae were counted with a manual counter and marked with a colored pen every 48 hours, using a different color to denote new pupae each day of observation. Once adults emerged, they were collected every 48 hours and counted. The numbers of eggs, pupae, and adults were used to assess stage-specific survival from egg to pupae and from pupae to adult. These counts were combined to establish overall survival from egg to adult. Survival to the pupal stage was determined by dividing the number of pupae formed by the number of eggs counted [39,40]. Pupation survival was determined by dividing the number of adults collected by the number of pupae counted [41,42]. Overall survival was determined by dividing the number of adults collected by the number of eggs counted. Percentages were used to quantify survival rates at each stage. All analyses were done in R v 4.4.2 [43].

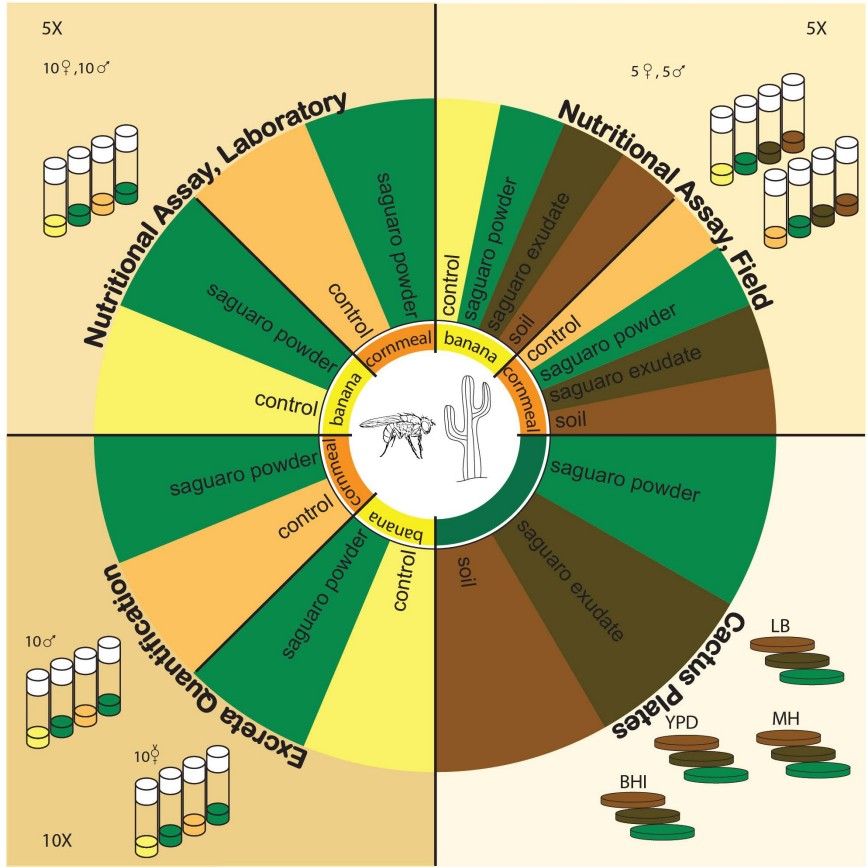

**Fig 1. Summary of experimental layout and methods of nutritional and behavioral assays.**

To quantify stage-specific and overall survival, we modeled survival as a binomial outcome, with the starting stage as the trial and successful progression to the next stage as a success. We used a generalized linear mixed-effects model (GLMM) with a binomial distribution from the *lme4* package in R [44,45]. Fixed effects included food type, cactus treatment, their interaction, and day of oviposition. Fly group was included as a random intercept to account for the multiple vials obtained from the same parental groups across days. A Type 3 ANOVA (from the *car* package) [46] was used to obtain significance tests for each of the categorical variables and interaction combinations. The *emmeans* package [47] was used to estimate corrected mean survival at each life stage on each diet.

## Excreta quantification assay

The Excreta Quantification (Ex-Q) Assay is adapted from Wu et. al [48], and was used to assess parental feeding on media with cactus present. The assay was conducted with four treatments: Banana media with and without cactus powder, and Cornmeal with and without cactus powder (Fig 1). Male and unmated female adult flies were evaluated in separate trials and aged separately on their treatments for three days before they were used for the experiment. Ex-Q tubes were created from 50 mL conical tubes with a hole cut on the top where a 1.5 mL centrifuge tube lid was placed and used as a food cup. The food cup was filled with 50 $\mu$L of dyed treatment food. Food was colored by mixing powder of the non-digestible food dye erioglaucine 1% (w/v). Each treatment had 5 replicates with 10 individuals in each vial. Flies were serially transferred to fresh Ex-Q tubes every 24 hours for two consecutive days, with excreta collected after each

24-hour feeding period. On the final day, flies were transferred to an empty vial for an additional 3-hour period to allow excretion of any remaining ingested dye, ensuring that consumption was captured through the final time point. Excreted dye was collected by adding 3 mL of deionized water to the Ex-Q tube, sealing the top opening, and spinning down the contents of the tube. Absorbance was measured at 630 nm in a 96-well plate on a spectrometer (Molecular Biosciences Spectra Max Series Plus 384). Each plate contained samples from all eight treatments over the total number of days. A standard curve was prepared by dissolving 10 mg of erioglaucine in 10 mL of PBS followed by a serial 2-fold dilution. The final range of standards was from 78 $\mu$g/mL to 5 $\mu$g/mL. A 100 $\mu$g/mL aliquot of standard (2 replicates) and treatments (3 replicate; 3 flies per treatment) were dispensed into individual wells of a 96-well plate. Each repeat was averaged for the final data analysis. The spectrophotometer absorbance readings were used to calculate the average amount of dye the flies excreted after feeding on the various diets. The collected dye therefore approximates how much the flies ate on each treatment based on the following equation:

$$\text{Dye } (\mu g) \text{ per fly} = \frac{(\text{Absorbance} - \text{Intercept}) \times \text{Dye collection volume}}{\text{Slope} \times \text{\# flies} \times \text{Aliquot volume}}$$

Normality of residuals was assessed using the Shapiro-Wilk test [49]. A Kruskal-Wallis rank sum test and a Dunn's post hoc test was conducted in R to examine diet consumption of male and female flies provided with media that contained cactus powder compared to controls for each food type [50–52].

## Cactus plates

Aliquots of Saguaro powder, exudate, and soil were cultured and plated to assess microbial community. Four types of microbial growth plates were used for this experiment: Yeast Peptone Dextrose (YPD), Luria Broth (LB) Agar, Brain Heart Infusion (BHI), and Mueller Hinton (MH). Three types of cactus samples were used: one control powder and two field samples (Fig 1). The Saguaro powder used for the Fitness Assay was frozen, defrosted, and opened multiple times. The two field samples were exudate and soil that was soaked in cactus exudate collected from Globe, Arizona. The three types of Saguaro will henceforth be called "treatments." Deionized water was used as a negative control. To plate, 14 mg of each treatment was suspended in 1 mL of deionized water and centrifuged for 10 seconds. The mixture was resuspended by pipetting up and down at least three times immediately prior to plating. A 40 $\mu$L aliquot of the suspension was plated on each type of plate (equating to about 56 $\mu$g of cactus added to each plate) and spread with glass beads. The plates were incubated at 21°C, which is the temperature at which the experiments using live *D. mettleri* were conducted. After 24 hours, images of each plate were taken. Plate pictures were processed using a plate count protocol in ImageJ to assess colony count and growth area on each medium [53]. Data were analyzed using a two-way analysis of variance (ANOVA) to assess the effects of cactus and plate type on the number, size, and total area of microbial colonies as a proxy for abundance [54]. Significance was determined based on the p-values, with Tukey's Honest Significant Difference (HSD) test used for post-hoc pairwise comparisons [55]. Normality of residuals was assessed using the Shapiro-Wilk test [49].

## Results

### Ex-Q Assay

We found no indication that the flies eat differently on any of the diet treatments, so any developmental differences should not arise due to parental consumption differences. Adults on Cornmeal showed a slight increase in the amount of dye excreted compared to adults on Banana media, but these results were not significant by a Kruskal-Wallis rank sum test (Figs 2, 3). Unmated males and females on food with cactus showed no significant difference in the amount of dye excreted when compared to respective controls (Fig 3).

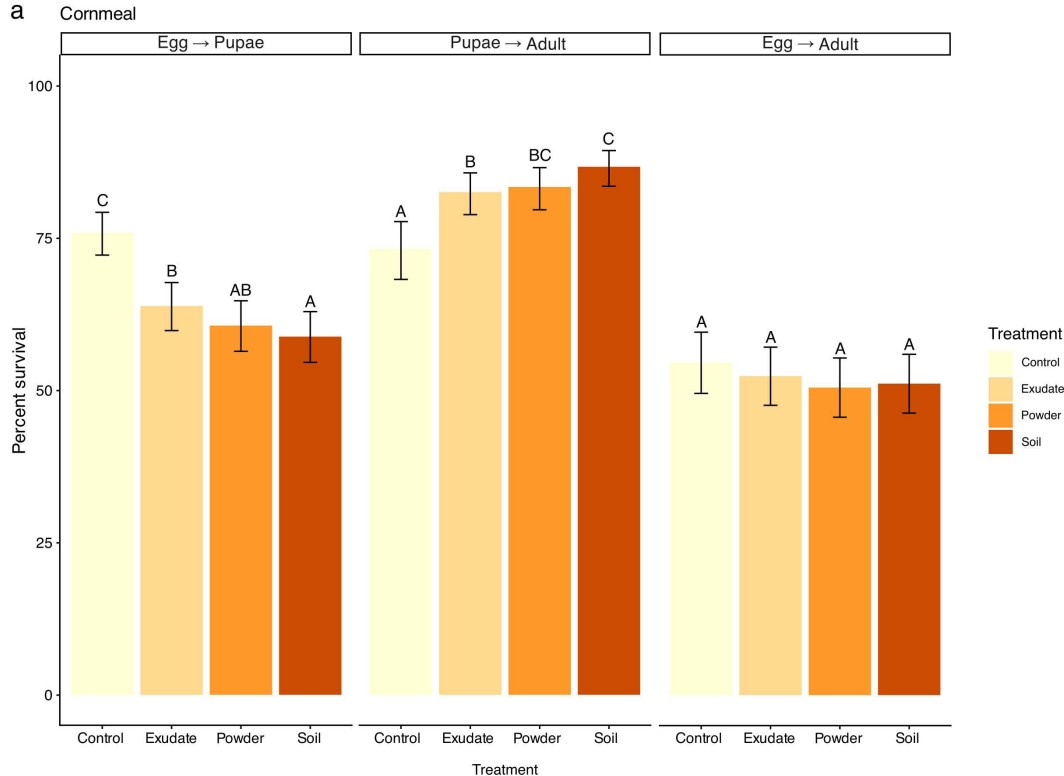

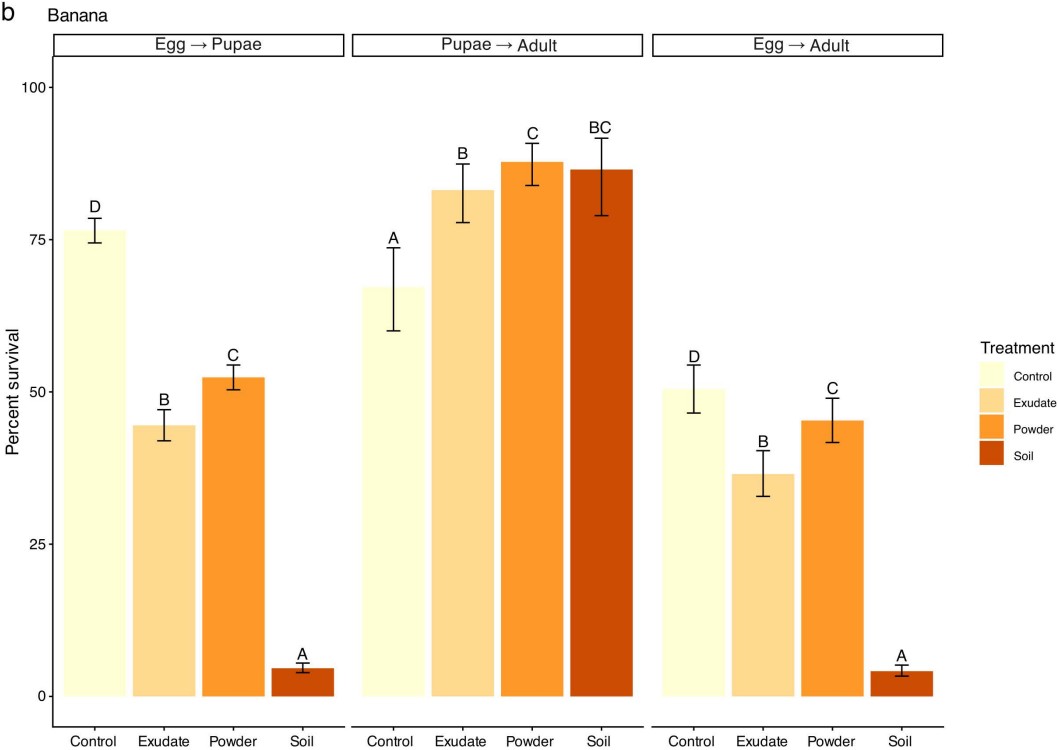

**Fig 2. Fitness Assay Results.** Mean (± 95% C.I.) Egg to pupae, pupae to adult, and cumulative egg to adult survival rates of *D. mettleri* on control (yellow), Saguaro exudate (tan), Saguaro powder (orange), and Saguaro soil (brown) on (a) Cornmeal and (b) Banana media, estimated as *emmeans* after accounting for experimental variables (see Supplemental Tables S2 and S3 Tables). Letters above error bars indicate difference between the means, with identical letters indicating results are not significantly different ($p < 0.05$) based on post-hoc contrast of *emmeans*.

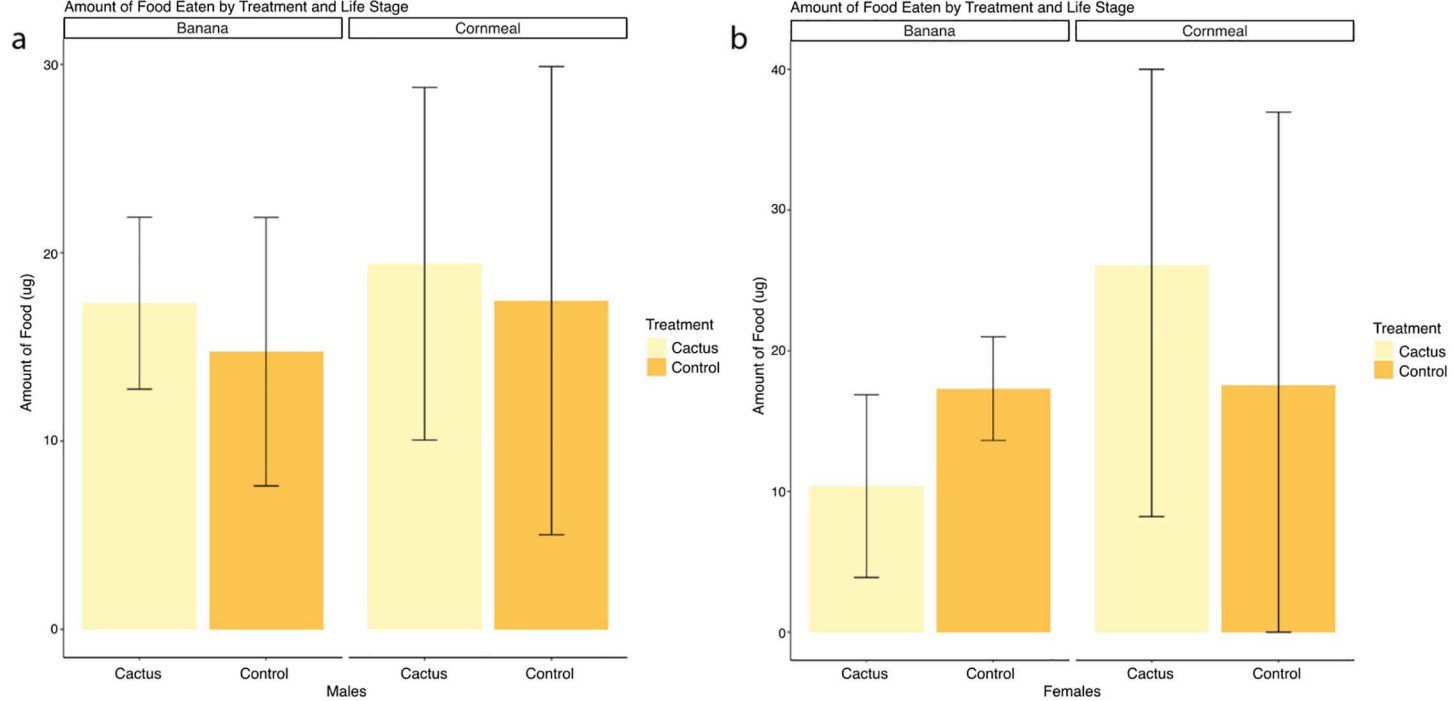

**Fig 3. Excrement Quantification Assay Results.** Mean (± 95% C.I.) amount of dye excreted in ug by males (a) and unmated females (b) on Banana and Cornmeal with Saguaro powder added and controls. Bars without letters are not significantly different based on Dunn's post hoc test ($p < 0.05$).

## Fitness assay

*Drosophila mettleri* survival differed across media type, cactus treatment, and life stage assessed (Fig 2, S1, S2, S3 Tables). Survival from egg to pupal stage was lower in all three cactus treatments than in controls (5–52% vs. 77% in Banana, 59–64% vs 76% in Cornmeal; Fig 2, S1 and S3 Tables). However, conditional on surviving to pupation, the percentage of individuals that continued to survive pupation and emerge as adults was significantly higher in cactus treatments than controls (83–88% vs 67% in Banana, 83–87% vs 73% in Cornmeal; Figs 2, S1 and S3 Tables). There were no significant stage-specific survival differences between most types of cactus additives for eclosion on Banana or all Cornmeal, except that pupal survival was higher on Banana-powder compared to Banana-exudate, and dramatically worse on the Banana-soil combination (Fig 2, S1 and S3 Tables).

Cumulative egg-to-adult survival differed significantly for Banana but not Cornmeal (Fig 2, S2, S3 Tables). On Banana media, cactus supplementation reduced survival relative to control food, with cactus soil having a strong negative effect. In contrast, cactus supplementation on Cornmeal did not significantly affect survival compared with control food (Fig 2, S2, S3 Tables). The duration from the first larvae to pupae and from pupae to adults was not significantly different across media (S1 Table). However, *Drosophila* reared on media containing Saguaro soil

## Microbial abundance

Saguaro powder, exudate, and soaked soil were plated on a variety of microbial culture media to determine whether viable yeast and bacteria were present (Fig 4). The Saguaro powder showed the lowest level of microbial activity, with either no or low (0–6.25 colonies) culture growth (Fig 4 and S4 Table). In contrast, Saguaro exudate and soil samples had more colonies (987–1710 colonies) and total area of growth (4164–4185 mm² vs 216 mm²), implying higher microbial activity

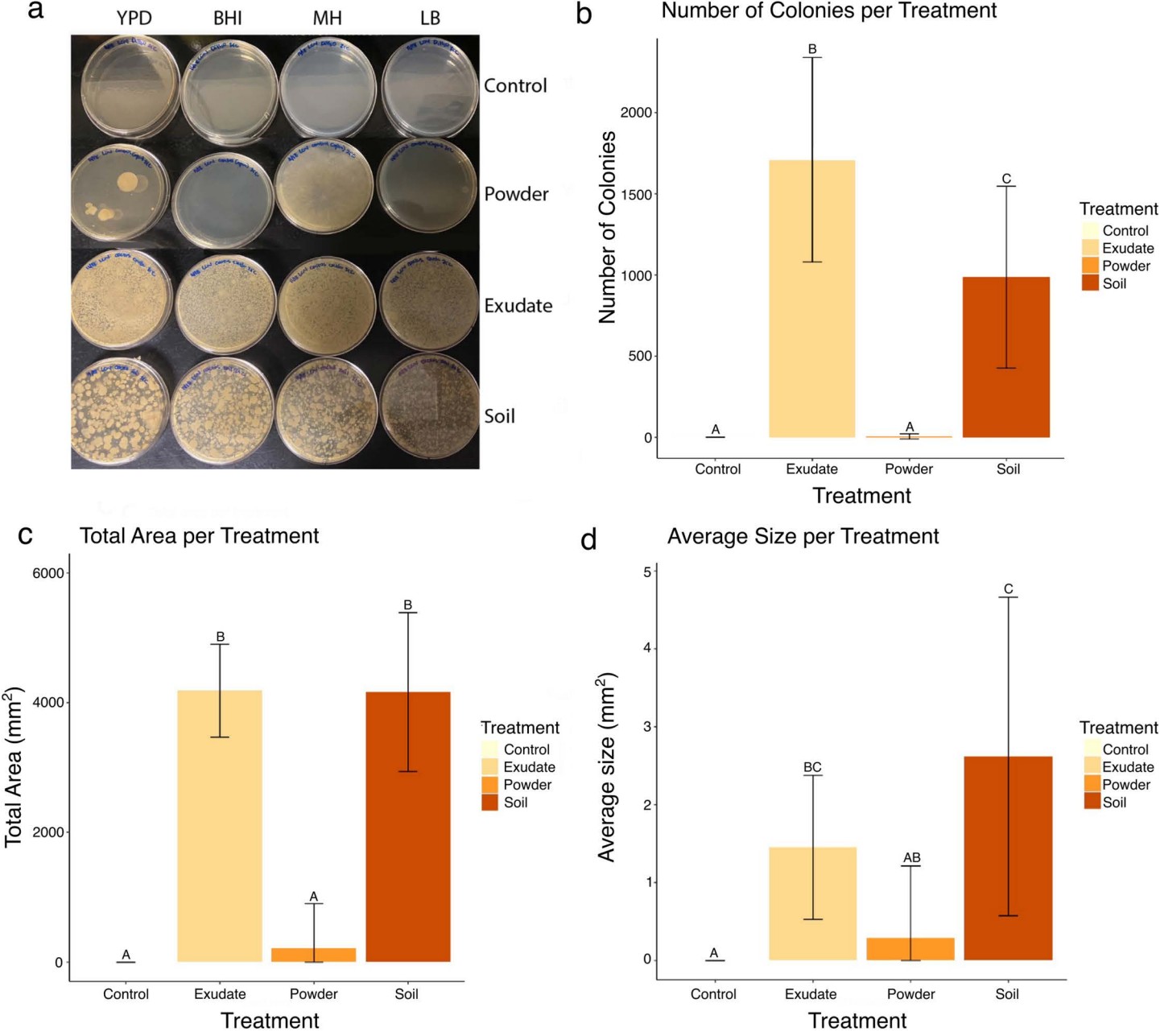

**Fig 4. Cactus Plates Results.** (a) YPD, BHI, MH, and LB plates depicting microbial growth of Control, Powder, Exudate, and Soil samples. Graphs of (b) Total Number of Colonies, (c) Total Colony Area, and (d) Average Colony Size per treatment of control (yellow), Saguaro exudate (tan), Saguaro powder (brown). Letters above error bars indicate difference between the means, with identical letters showing not significantly different based on Tukey's HSD post hoc test ($p<0.05$).

than plates with powder samples (Fig 4 and S4 Table). While plates with Saguaro soil had colony sizes twice as large as exudate samples, Saguaro exudate had almost twice as many microbial colonies potentially indicating different microbial abundance between the two additives or that Saguaro exudate has higher microbial activity relative to Saguaro soil (Fig 4 and S4 Table).

## Discussion

This study evaluated how various Saguaro preparations and media types affect *D. mettleri* stage-specific survival. We found that cactus supplementation had opposing effects across developmental stages, reducing early stage survival, but increasing survival from pupation to adulthood (Fig 2 and S1 Table). Cactus supplementation had an overall negative effect on egg-to-adult survival, therefore lowering fitness. Egg-to-adult development reflects the consequences of parental oviposition decisions, since parental flies select the egg-laying environment. Because adult food intake did not differ among treatments, parental feeding differences are unlikely to account for the observed variation in offspring survival (Fig 3). These results suggest that cactus- derived substrates alter the developmental environment in ways that influence survival, potentially through changes in nutrient availability, microbial activity, or their interaction. These results are in line with other studies on specialist *Drosophila* species, where *mojavensis*, *elegans*, and *sechellia* larvae had lower body weights, higher mortality, and delayed development when compared to *melanogaster* and/or *simulans* on high sugar, lower protein diets [27,56].

Fitness can be influenced by the environment of the fly, including by larval competition [28,57–59]. In the wild, the adults lay hundreds of eggs on a suitable larval development substrate. Once the eggs hatch, the hundreds of larvae begin to compete with one another for resources. Larvae with an earlier hatching date gain precedence on resources within the environment, influencing success in the later stages. While the larvae have more substrate to explore in the wild, they are confined to a certain amount of food and space within a tube under lab conditions. The finite resources lead to heightened larval competition, resulting in potential delayed development, reduced size, or increased mortality [57–59]. The addition of powdered cactus and microbially active exudate adds another layer of complexity by changing the environment. The cactus boosts the nutritional value of the media by providing sterols and simple chemicals that lab media generally [15,16]. These additions can promote faster growth and shorter development times, which explains the development patterns observed in flies reared on Saguaro powder and exudate (S1 Table) [25–27,29].

Depending on the interplay between a poor diet and heightened larval competition, improving the nutritional quality of the media may not be beneficial to *D. mettleri* fitness [59]. Some soil microbes can enhance nutrient availability by either breaking down cactus into more bioavailable constituents or by serving as a nutrient source for the *Drosophila* [32,34,60–62]. At the same time, some of these microbes can consume nutrients in the substrate and may proliferate faster than larvae can develop, such that the microbes overtake the environment and compete with larvae for resources [57,58]. These factors collectively impact individual fitness.

The Cactus Plates assay (Fig 4) explores the microbial abundance in the different treatments. Cactus powder had the lowest microbial activity, Saguaro exudate had the highest, and Saguaro soil was intermediate in activity (S4 Table). Lower microbial activity in the cactus powder is expected since it underwent extreme conditions during its manufacturing and storage, probably limiting the microbial community that could survive. The soil treatment included Saguaro exudate that would have leached into the soil in addition to microbes that may have been present in the cactus and/or soil. However, the number of microbial colonies recovered from soil was about half of what was recovered from the exudate, suggesting that not all microbes present in cactus exudate survive in the soil. Nevertheless, soil-derived substrates influenced development time and survival outcomes in our study. Whether these effects arise from microbial activity, substrate chemistry, or both cannot be resolved from our data but could be a subject for future study.

The amount of media consumed by flies on the Cornmeal diet compared to the Banana diet did not differ for unmated females or males and is not consistent with the results of similar studies conducted on other species [48]. According to the *Drosophila* Dietary Composition Calculator (DDCC), the Cornmeal diet with no cactus additions has a protein: carbohydrate ratio (P:C) of 1:3.5 [34]. The Banana diet with no cactus additions has a protein: carbohydrate ratio of 1:11, indicating this diet has 3 times more carbohydrates than the Cornmeal diet. We added 14 mg/100g of cactus powder, which adds 4.69 mg/100g of protein and 1.44 mg/100g of carbohydrates, significantly increasing the protein: carbohydrate ratios to 1:3 and 1:8 for Cornmeal and Banana [63]. Adult *Drosophila melanogaster* fed on a high sugar diet (> 8% w/v) consume

a significantly lower amount than those fed on a low sugar diet (< 4% w/v) [35,64]. Our present data suggest that, unlike *D. melanogaster*, *D. mettleri* do not exhibit feeding differences based on the amount of sugar (or other Saguaro macronutrients) in the diet. This may reflect dramatic differences in the natural ecology of the two flies, as *D. melanogaster* feeds and develops on sugar-rich fermenting fruit in nature [9,20].

Our experimental design did not allow for interaction between unmated females and males. The results we observe may have been biased because females in the absence of males may have more time to feed, or because mated females may increase feeding relative to unmated females because of the nutritional requirements of egg production. Mated *Drosophila melanogaster* females are more likely to make nutritional choices based on the ovipositional site that will provide their progeny with the best fitness outcomes [26,28,29,65]. In *D. melanogaster*, unmated females and males tend to consume more carbohydrates in their diet (P:C 1:4) while mated females tend to consume comparatively more protein (P:C 1:1.5) [25,65]. While *D. melanogaster* and *D. mettleri* are taxonomically distant, their nutritional requirements could be evolutionarily conserved as a function of the need to provision eggs.

Our results demonstrate that Saguaro does not uniformly affect *Drosophila mettleri* fitness under lab conditions, but instead has context- and life stage-dependent effects. Although cactus supplementation reduced survival prior to pupation, it increased survival during pupation, and the effect of cactus supplementation on overall egg-to-adult outcomes depended critically on the base diet. In particular, cactus soil was highly detrimental on banana-based food but far less harmful on cornmeal-based food, indicating that the broader nutritional environment modulates how cactus-derived substrates affect development. In a resource-limited environment of the Sonoran Desert, such a dependency can be a competitive advantage by enabling *D. mettleri* to exploit a niche that is inaccessible to other organisms. More broadly, our study highlights how host plant chemistry can directly shape fitness in specialist insects, offering insight into the mechanisms underlying host-use evolution and the persistence of specialization under extreme environmental pressures. These results add to the growing field of nutritional ecology and lay the groundwork for understanding physiological processes leading to *D. mettleri*'s adaptation to Saguaro cactus.

## Conclusion

Our results show that Saguaro additives have stage-specific and context-dependent effects on *Drosophila mettleri* fitness. Rather than producing uniform effects across development, Saguaro components differentially affected survival and performance across life stages and dietary conditions. These findings suggest that host-associated fitness consequences are not fixed properties of a resource but instead emerge from interactions between developmental stage and environmental context. In extreme environments such as the Sonoran Desert, such conditional relationships can shape how organisms respond to variable resource landscapes. More broadly, our results contribute to a growing understanding of nutritional and environmental factors interact to structure host-associated performance in specialist insects and provides a framework for future work to disentangle the chemical, nutritional, and microbial mechanisms underlying these effects.

## Supporting information

**S1 Table. Table of mean survival and development times (days).** Table shows egg, pupa, and adult *Drosophila mettleri* results for the nutritional assay shown in Fig 1.
(PDF)

**S2 Table. Table of ANOVA results.** Type III ANOVA from generalized linear mixed models (GLMMs) testing the effects of cactus treatment and day on survival across developmental stages and diet types (cornmeal and banana). Significance codes: ***$p < 0.001$, **$p < 0.01$, *$p < 0.05$.
(PDF)

**S3 Table. Table of post-hoc results.** Estimated survival probabilities (%) across developmental stages (egg-to-adult, egg-to-pupae, pupae-to-adult) for each cactus treatment and diet type (cornmeal and banana).
(PDF)

**S4 Table. Table of agar plate readings.** Count indicates the number of colonies found. Total area refers to the total area of microbial coverage. Average size refers to the average size of colonies. %Area indicates the percentage colony cover compared to the total area. Mean refers to the average pixel value.
(PDF)

## Acknowledgments

We thank Saguaro National Park (permit SAGU-2023-SCI-0007) and park rangers for their support throughout the study. We thank Benjamin Burgunder, Andrew Legan, Alan Mata, Kyla O'Hearn, and Augusto Santos Rampasso for their assistance in the field.

## Author contributions

**Conceptualization:** Lidane Noronha, Brian P. Lazzaro, Patrick M. O'Grady.

**Data curation:** Lidane Noronha.

**Formal analysis:** Lidane Noronha.

**Funding acquisition:** Patrick M. O'Grady.

**Investigation:** Lidane Noronha.

**Methodology:** Lidane Noronha, Brian P. Lazzaro, Patrick M. O'Grady.

**Project administration:** Lidane Noronha.

**Resources:** Lidane Noronha.

**Software:** Lidane Noronha.

**Supervision:** Brian P. Lazzaro, Patrick M. O'Grady.

**Validation:** Lidane Noronha, Brian P. Lazzaro, Patrick M. O'Grady.

**Visualization:** Lidane Noronha.

**Writing – original draft:** Lidane Noronha.

**Writing – review & editing:** Brian P. Lazzaro, Patrick M. O'Grady.

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
