## [Decision Letter · Decision Letter 0]

1 Dec 2025

PONE-D-25-48608Host plant nutrition drives fitness outcomes in the cactus specialist Drosophila mettleriPLOS ONE

Dear Dr. Noronha,

Thank you for submitting your manuscript to PLOS ONE. After careful consideration, we feel that it has merit but does not fully meet PLOS ONE’s publication criteria as it currently stands. Therefore, we invite you to submit a revised version of the manuscript that addresses the points raised during the review process.

Please note that reviewer made a careful analysis of your work and raised several criticisms. However, It seems to me that most of them can be adjusted by a reinterpretation/rewriting (comments #1,2,3 and 6) or re-analysis (comment #5) of the data. In comment #4, they asked for additional data that you possibly have.

We look forward to receiving your revised manuscript.

Kind regards,

Pedro L. Oliveira

Academic Editor

PLOS ONE

Journal Requirements:

[This paper was supported by National Science Foundation (https://www.nsf.gov/bio/deb) grants DEB2030129, DEB1839598, and DEB1241253 to PMO.

The Cornell University Insect Collection (https://cuic.entomology.cornell.edu/) and the American Museum of Natural History Theodore Roosevelt Memorial Grant (https://www.amnh.org/research/richard-gilder-graduate-school/grants) provided financial support to LN.].

4. Please amend the manuscript submission data (via Edit Submission) to include author Lidane Noronha.

5. Please amend your authorship list in your manuscript file to include author Lidane Audrey Noronha.

7. Please upload a new copy of Figures 2 and 3 as the details are not clear. Please follow the link for more information:  https://journals.plos.org/plosone/s/figures

Reviewers' comments:

Reviewer's Responses to Questions

**Comments to the Author**

1. Is the manuscript technically sound, and do the data support the conclusions?

Reviewer #1: Partly

2. Has the statistical analysis been performed appropriately and rigorously? 

Reviewer #1: No

3. Have the authors made all data underlying the findings in their manuscript fully available?

Reviewer #1: Yes

4. Is the manuscript presented in an intelligible fashion and written in standard English?

Reviewer #1: Yes

5. Review Comments to the Author

Reviewer #1: In my review I considered the seven editorial criteria required for acceptance in PLoS One. My comments below mainly concern criterion 3 (Experiments, statistics, and other analyses are performed to a high technical standard and are described in sufficient detail) and criterion 4 (Conclusions are presented in an appropriate fashion and are supported by the data). Although I believe the manuscript does not meet these criteria in its current form, the issues I note below could be fixed in a revision by providing some clarifications, redoing some analyses, and noting caveats to interpretations of the results. In my opinion, the paper meets the other five criteria.

1. Drosophila mettleri specialization: throughout the paper, the authors treat D. mettleri as a strict saguaro specialist. For example, lines 9-10 from the abstract. But this is not the case as populations of D. mettleri are found in areas where saguaro cactus does not occur (e.g. Santa Catalina Island and Baja peninsula).

2. If I understood correctly, the line used in the study was actually collected from the Baja peninsula. Saguaro cactus is not present there; the main host is cardon. I think the authors need to acknowledge this as a caveat to the interpretation of their results. Furthermore, the line has been in the lab since 2012, which means it may be somewhat lab adapted. Again, this should be noted as a caveat.

3. Food formulations: The authors should provide more detail about how the food treatments were prepared. For example, I am unclear on exactly what the soil treatment consisted of. How was it collected, how was it stored, how much was applied (that goes for all treatments)?

4. Fitness assay: the most relevant measure of fitness in this assay is egg to adult survival, but the authors do not report this information. Instead, they analyze egg to pupa survival and pupa to adult survival as separate measures. I think these are reasonable measures to include for a more nuanced look, but ultimately fitness depends on the combined effects of both survival rates so this should be the main analysis. From looking at the data, this looks like it could change some conclusions or shift the interpretation, because it appears there might be a tradeoff between survival from egg to pupa vs pupa to adult. I would also suggest the authors not describe this as a “nutritional assay.” While there are no doubt nutrient differences between their treatments there are numerous other differences as well (e.g. toxicity, etc.). I don’t really think the experimental design allows you isolate the effects of nutritional differences specifically. There are also some contradictory statements—e.g. in the abstract it says microbial communities appear to improve developmental success (lines 12-13), while in the discussion it says that it is the nutritional properties of the host plant that are most critical (lines 318-319). Basically, I don’t think the design permits conclusions as to which is more important, or if it does, this needs to be better explained.

5. Statistical analysis of fitness assay: If I understand the experimental design correctly, the egg counts for each group of five pairs of flies comes from multiple separate vials, not just one. The fact that they have repeated measurements from a single group is not accounted for in their statistical analysis. This problem could be remedied with a different statistical approach. I would recommend a GLMM with binomial response that includes “fly group” as a random effect. This approach would deal with the repeated measures and the GLMM approach would likely provide much more power than the Kruskall-Wallis test.

6. I am not familiar with the excreta assay. Could the authors provide a bit more detail? For example, I am struggling to understand why the excreta is only measured for 3 hours at the end of several days of feeding. What if they excrete a bunch of dye right before you transfer them to the last vial? Or what if they don’t excrete in the 3 hours even though they ate dye? Some more clarifications/explanations on these points would be helpful. Also lines 194-196 they say “any developmental differences did not arise due to consumption differences.” But consumption was measured on adults, while by development I assume they mean egg to adult. It is not clear how these are connected.

7. Microbial assays: The authors discuss their results in terms of microbial diversity, but it is not clear to me that they measured diversity per se (i.e. the number of different types of bacteria/yeasts). It seems like they quantified biomass. More clarification or a change in the wording is needed. Also, they should briefly mention the caveat that these assays are only relevant for microbes that are easily cultured with the different media they used. There could be hidden diversity that does not culture well.

8. References:

a. The authors did not include one reference that is quite pertinent to their study and might help frame some expectations for differences between laboratory and cactus food: Hoang K, Matzkin LM, Bono JM. Transcriptional variation associated with cactus host plant adaptation in Drosophila mettleri populations. Mol Ecol. 2015 Oct;24(20):5186-99. doi: 10.1111/mec.13388. Epub 2015 Oct 12. PMID: 26384860.

b. Lines 58-60: the wording makes it sound as though these references show this effect specifically for saguaro, but these are in fact studies from different host plants.

6. PLOS authors have the option to publish the peer review history of their article (what does this mean?). If published, this will include your full peer review and any attached files.

Reviewer #1: No

---

## [Author Response · Author response to Decision Letter 1]

19 Feb 2026

Review

We thank the reviewer and editor for their thoughtful and constructive feedback. Below, we provide a point-by-point response. Journal and reviewer comments are in black, followed by author responses in green. Line numbers refer to “Revised manuscript (clean copy)”.

Journal Requirements:

We have reformatted the manuscript, updated figure and table titles, and corrected file naming to meet PLOS ONE style guidelines.

We have added full permit information in the Methods (lines 95 – 97), including the authority granting site access.

[This paper was supported by National Science Foundation (https://www.nsf.gov/bio/deb) grants DEB2030129, DEB1839598, and DEB1241253 to PMO.

The Cornell University Insect Collection (https://cuic.entomology.cornell.edu/) and the American Museum of Natural History Theodore Roosevelt Memorial Grant (https://www.amnh.org/research/richard-gilder-graduate-school/grants) provided financial support to LN.].

We added all sources of financial support, including internal support. The revised statement will include:

“There was no additional external funding received for this study.”

4. Please amend the manuscript submission data (via Edit Submission) to include author Lidane Noronha.

The manuscript submission data were updated accordingly to list the author as Lidane Noronha.

5. Please amend your authorship list in your manuscript file to include author Lidane Audrey Noronha.

The manuscript file was updated to ensure consistency by listing the author as Lidane Noronha.

We clarified this statement in lines 93 – 95.

7. Please upload a new copy of Figures 2 and 3 as the details are not clear. Please follow the link for more information: https://journals.plos.org/plosone/s/figures

We have uploaded new copies of Figures 2 and 3 with improved details.

Reviewers' comments:

Reviewer's Responses to Questions

Comments to the Author

1. Is the manuscript technically sound, and do the data support the conclusions?

Reviewer #1: Partly

2. Has the statistical analysis been performed appropriately and rigorously?

Reviewer #1: No

3. Have the authors made all data underlying the findings in their manuscript fully available?

Reviewer #1: Yes

4. Is the manuscript presented in an intelligible fashion and written in standard English?

Reviewer #1: Yes

5. Review Comments to the Author

Reviewer #1: In my review I considered the seven editorial criteria required for acceptance in PLoS One. My comments below mainly concern criterion 3 (Experiments, statistics, and other analyses are performed to a high technical standard and are described in sufficient detail) and criterion 4 (Conclusions are presented in an appropriate fashion and are supported by the data). Although I believe the manuscript does not meet these criteria in its current form, the issues I note below could be fixed in a revision by providing some clarifications, redoing some analyses, and noting caveats to interpretations of the results. In my opinion, the paper meets the other five criteria.

1. Drosophila mettleri specialization: throughout the paper, the authors treat D. mettleri as a strict saguaro specialist. For example, lines 9-10 from the abstract. But this is not the case as populations of D. mettleri are found in areas where saguaro cactus does not occur (e.g. Santa Catalina Island and Baja peninsula).

We thank the reviewer for suggesting this clarification. We have revised the manuscript to avoid overstating specialization on saguaro cactus. Throughout the text, we now refer to D. mettleri as being associated with columnar cacti in general (lines 18 – 19, 48 – 50) and clarify that the population used in this study was collected in Baja California, where Saguaro is absent, and cardón (Pachycereus pringlei) is the primary host (line 84).

2. If I understood correctly, the line used in the study was actually collected from the Baja peninsula. Saguaro cactus is not present there; the main host is cardon. I think the authors need to acknowledge this as a caveat to the interpretation of their results. Furthermore, the line has been in the lab since 2012, which means it may be somewhat lab adapted. Again, this should be noted as a caveat.

As noted above, we mention that the cardón is the main host in the geographic region where the studied D. mettleri strain was originally collected (line 84). We also note that this line has been maintained in the laboratory for several years, and that the strain may exhibit a degree of laboratory adaptation (lines 85 – 87).

3. Food formulations: The authors should provide more detail about how the food treatments were prepared. For example, I am unclear on exactly what the soil treatment consisted of. How was it collected, how was it stored, how much was applied (that goes for all treatments)?

We have expanded the Methods section to clearly describe the collection, preparation, storage, and concentration of cactus-derived material used in the food treatments (lines 88 – 93).

4. Fitness assay: the most relevant measure of fitness in this assay is egg to adult survival, but the authors do not report this information. Instead, they analyze egg to pupa survival and pupa to adult survival as separate measures. I think these are reasonable measures to include for a more nuanced look, but ultimately fitness depends on the combined effects of both survival rates so this should be the main analysis. From looking at the data, this looks like it could change some conclusions or shift the interpretation, because it appears there might be a tradeoff between survival from egg to pupa vs pupa to adult. I would also suggest the authors not describe this as a “nutritional assay.” While there are no doubt nutrient differences between their treatments there are numerous other differences as well (e.g. toxicity, etc.). I don’t really think the experimental design allows you isolate the effects of nutritional differences specifically. There are also some contradictory statements—e.g. in the abstract it says microbial communities appear to improve developmental success (lines 12-13), while in the discussion it says that it is the nutritional properties of the host plant that are most critical (lines 318-319). Basically, I don’t think the design permits conclusions as to which is more important, or if it does, this needs to be better explained.

We thank the reviewer for this suggestion and agree that egg-to-adult survival provides a more comprehensive fitness proxy. We have reanalyzed the data using egg-to-adult survival and updated the Results and Discussion accordingly (lines 132 – 138, 258 - 260). This change strengthens the biological interpretation of our findings. We have retained the separate analysis of survival and development time from egg-to-pupa and pupa-to-adult in Figure 2 and as a supplemental Table S1 for additional detail and resolution.

5. Statistical analysis of fitness assay: If I understand the experimental design correctly, the egg counts for each group of five pairs of flies comes from multiple separate vials, not just one. The fact that they have repeated measurements from a single group is not accounted for in their statistical analysis. This problem could be remedied with a different statistical approach. I would recommend a GLMM with binomial response that includes “fly group” as a random effect. This approach would deal with the repeated measures and the GLMM approach would likely provide much more power than the Kruskall-Wallis test.

We agree and have revised the statistical analysis to explicitly account for the observation of multiple vials originating from the same groups of flies. We now analyze survival using a binomial generalized linear mixed model that includes fly group as a random effect (lines 136 – 138, results in Table 1). This model structure appropriately reflects the experimental design and resolves concerns about non-independence among vials derived from the same group of flies.

6. I am not familiar with the excreta assay. Could the authors provide a bit more detail? For example, I am struggling to understand why the excreta is only measured for 3 hours at the end of several days of feeding. What if they excrete a bunch of dye right before you transfer them to the last vial? Or what if they don’t excrete in the 3 hours even though they ate dye? Some more clarifications/explanations on these points would be helpful. Also lines 194-196 they say “any developmental differences did not arise due to consumption differences.” But consumption was measured on adults, while by development, I assume they mean egg to adult. It is not clear how these are connected.

We have clarified the timing and interpretation of the excreta assay in the Methods (lines 152 – 156) and Discussion (lines). The assay reflects adult feeding behavior during a defined observation window and does not measure larval consumption or developmental diet preference. Adult consumption in our study reflects parental feeding, which shapes the developmental environment of offspring (lines 144 – 145). We now state this explicitly to avoid misinterpretation.

To run the excreta quantification assay, flies were transferred to fresh Ex-Q tubes every 24 hours over two consecutive days, and excreta were collected after each 24-hour feeding period (lines 152 – 154). The final 3-hour transfer to an empty vial was included specifically to allow clearance of any dye remaining in the gut, ensuring that dye ingested during the final feeding interval was captured. Thus, total dye recovery reflects cumulative excretion across the full feeding period rather than a single 3-hour window.

7. Microbial assays: The authors discuss their results in terms of microbial diversity, but it is not clear to me that they measured diversity per se (i.e. the number of different types of bacteria/yeasts). It seems like they quantified biomass. More clarification or a change in the wording is needed. Also, they should briefly mention the caveat that these assays are only relevant for microbes that are easily cultured with the different media they used. There could be hidden diversity that does not culture well.

We have revised the manuscript (lines 71 – 73, 99 – 100, 187 – 189, 143 – 146, 239 – 246, 262 – 264, 288 – 292, 297 – 299) to avoid implying quantification of microbial diversity in the ecological sense. However, our measure is also not a measure of biomass, in the sense that we have not quantified microbial abundance within flies or substrates. Our measure is microbial abundance as quantified by colony count and growth area (186 – 189). The text now consistently refers to microbial abundance or activity, which most accurately reflects the scope of the measurements conducted in this study.

8. References:

a. The authors did not include one reference that is quite pertinent to their study and might help frame some expectations for differences between laboratory and cactus food: Hoang K, Matzkin LM, Bono JM. Transcriptional variation associated with cactus host plant adaptation in Drosophila mettleri populations. Mol Ecol. 2015 Oct;24(20):5186-99. doi: 10.1111/mec.13388. Epub 2015 Oct 12. PMID: 26384860.

b. Lines 58-60: the wording makes it sound as though these references show this effect specifically for saguaro, but these are in fact studies from different host plants.

We thank the reviewer for this suggestion. The study by Hoang et al. (2015) is now included (citation 21).

6. PLOS authors have the option to publish the peer review history of their article (what does this mean?). If published, this will include your full peer review and any attached files.

Do you want your identity to be public for this peer review? For information about this choice, including consent withdrawal, please see our Privacy Policy.

Reviewer #1: No

---

## [Decision Letter · Decision Letter 1]

19 Mar 2026

PONE-D-25-48608R1

Host plant nutrition drives fitness outcomes in the cactus specialist Drosophilamettleri

PLOS One

Dear Dr. Noronha,

Thank you for submitting your manuscript to PLOS ONE. After careful consideration, we feel that it has merit but does not fully meet PLOS ONE’s publication criteria as it currently stands. Therefore, we invite you to submit a revised version of the manuscript that addresses the points raised during the review process.

We look forward to receiving your revised manuscript.

Kind regards,

Pedro L. Oliveira

Academic Editor

PLOS One

Journal Requirements:

Additional Editor Comments:

This time the reviewer has focus don three specific points. While comments 2 and 3 are very simple details to be fixed, comment #1 is still about one of the major points raised in the first round, and need attention.

Reviewer's Responses to Questions

**Comments to the Author**

1. If the authors have adequately addressed your comments raised in a previous round of review and you feel that this manuscript is now acceptable for publication, you may indicate that here to bypass the “Comments to the Author” section, enter your conflict of interest statement in the “Confidential to Editor” section, and submit your "Accept" recommendation.

Reviewer #1: (No Response)

2. Is the manuscript technically sound, and do the data support the conclusions?

Reviewer #1: Partly

3. Has the statistical analysis been performed appropriately and rigorously? 

Reviewer #1: No

4. Have the authors made all data underlying the findings in their manuscript fully available?

Reviewer #1: Yes

5. Is the manuscript presented in an intelligible fashion and written in standard English?

Reviewer #1: Yes

6. Review Comments to the Author

Reviewer #1: I appreciate that the author’s attempted to address my previous comments, but I still have a some concerns that I think need to be further addressed.

1. The authors conducted the GLMM for egg to adult survival as suggested. However, they still treat this as secondary to the egg to pupae, and pupae to adult analyses. It seems to me that it makes the most sense to start with egg-adult as the overall measure of fitness and then deconstruct it into stage specific analyses if they want to highlight how survival varies across developmental stage. Most importantly, the stage-specific analyses should also utilize GLMM instead of the Kruskall-Wallis test. A final point: the authors presented the results of the GLMM as a series of parameter estimates. I think they should consider summarizing the analysis with an anova table (e.g. the ‘car’ package can be used to produce an anova table of GLMM) and potential post hoc follow-up tests as this may be more easily digestible by readers. They may also consider analyzing the banana and cornmeal experiments separately.

2. I do not see a figure 4 although it is referenced in the text.

3. Lines 258-260: I do not understand how egg to adult survival reflects the consequences of parental feeding behavior. Perhaps it is meant to be parental oviposition behavior?

7. PLOS authors have the option to publish the peer review history of their article (what does this mean?). If published, this will include your full peer review and any attached files.

Reviewer #1: No

---

## [Author Response · Author response to Decision Letter 2]

15 Apr 2026

Reviewer #1: I appreciate that the author’s attempted to address my previous comments, but I still have a some concerns that I think need to be further addressed.

The authors conducted the GLMM for egg to adult survival as suggested. However, they still treat this as secondary to the egg to pupae, and pupae to adult analyses. It seems to me that it makes the most sense to start with egg-adult as the overall measure of fitness and then deconstruct it into stage specific analyses if they want to highlight how survival varies across developmental stage. Most importantly, the stage-specific analyses should also utilize GLMM instead of the Kruskall-Wallis test. A final point: the authors presented the results of the GLMM as a series of parameter estimates. I think they should consider summarizing the analysis with an anova table (e.g. the ‘car’ package can be used to produce an anova table of GLMM) and potential post hoc follow-up tests as this may be more easily digestible by readers. They may also consider analyzing the banana and cornmeal experiments separately.

We thank the reviewer for these constructive suggestions and have revised the manuscript accordingly in several ways.

We now clarify that all survival analyses (egg-to-pupae, pupae-to-adult, and cumulative egg-to-adult) were conducted using a generalized linear mixed-effects model (GLMM) with a binomial distribution, ensuring a consistent analytical framework across life stages. Cornmeal and banana responses were analyzed separately to assess diet related survival. To improve interpretability, we report results using Type III ANOVA tables (via the car package) along with estimated marginal means and post hoc contrasts (via emmeans), rather than focusing on individual parameter estimates (see Fig. 2 and Supporting Tables S2-S3).

We also appreciate the suggestion to emphasize egg-to-adult survival as the overall measure of fitness. In the revised manuscript, we have clarified the presentation to more explicitly link stage-specific results to cumulative egg-to-adult outcomes. While stage-specific survival is presented to highlight differences across developmental stages, these results are integrated within the same figure and Discussion section to emphasize their contribution to overall fitness.

2. I do not see a figure 4 although it is referenced in the text.

We thank the reviewer for catching this omission. Figure 4 has now been included in the revised manuscript materials and is properly referenced in the Results section.

3. Lines 258-260: I do not understand how egg to adult survival reflects the consequences of parental feeding behavior. Perhaps it is meant to be parental oviposition behavior?

We agree that egg-to-adult survival is more directly linked to parental oviposition behavior, as adults determine the developmental environment experienced by their offspring. We have revised the text accordingly to reflect this interpretation (lines 252-253).

---

## [Editor Report · Decision Letter 2]

20 Apr 2026

Host plant nutrition drives fitness outcomes in the cactus specialist Drosophila mettleri

PONE-D-25-48608R2

Dear Dr. Noronha,

We’re pleased to inform you that your manuscript has been judged scientifically suitable for publication and will be formally accepted for publication once it meets all outstanding technical requirements.

Kind regards,

Pedro L. Oliveira

Academic Editor

PLOS One
---

## [Editor Report · Acceptance letter]

PONE-D-25-48608R2

PLOS One

Dear Dr. Noronha,

I'm pleased to inform you that your manuscript has been deemed suitable for publication in PLOS One. Congratulations! Your manuscript is now being handed over to our production team.

Kind regards,

on behalf of

Dr. Pedro L. Oliveira

Academic Editor

PLOS One